# The Functional Characterization of Carboxylesterases Involved in the Degradation of Volatile Esters Produced in Strawberry Fruits

**DOI:** 10.3390/ijms24010383

**Published:** 2022-12-26

**Authors:** Lingjie Zhang, Kang Zhou, Maohao Wang, Rui Li, Xinlong Dai, Yajun Liu, Xiaolan Jiang, Tao Xia, Liping Gao

**Affiliations:** 1School of Life Sciences, Anhui Agricultural University, Hefei 230036, China; 2State Key Laboratory of Tea Plant Biology and Utilization, Anhui Agricultural University, Hefei 230036, China; 3College of Tea Science, Guizhou University, Guiyang 550025, China

**Keywords:** strawberry, FaCXEs, volatile ester, carboxylesterases

## Abstract

Volatile ester compounds are important contributors to the flavor of strawberry, which affect consumer preference. Here, the GC-MS results showed that volatile esters are the basic aroma components of strawberry, banana, apple, pear, and peach, and the volatile esters were significantly accumulated with the maturation of strawberry fruits. The main purpose of this study is to discuss the relationship between carboxylesterases (CXEs) and the accumulation of volatile ester components in strawberries. FaCXE2 and FaCXE3 were found to have the activity of hydrolyzing hexyl acetate, *Z*-3-hexenyl acetate, and *E*-2-hexenyl acetate to the corresponding alcohols. The enzyme kinetics results showed that FaCXE3 had the higher affinity for hexyl acetate, *E*-2-hexenyl acetate, and *Z*-3-hexenyl acetate compared with FaCXE2. The volatile esters were mainly accumulated at the maturity stages in strawberry fruits, less at the early stages, and the least during the following maturation stages. The expression of FaCXE2 gradually increased with fruit ripening and the expression level of FaCXE3 showed a decreasing trend, which suggested the complexity of the true function of CXEs. The transient expression of *FaCXE2* and *FaCXE3* genes in strawberry fruits resulted in a significantly decreased content of volatile esters, such as *Z*-3-hexenyl acetate, methyl hexanoate, methyl butyrate, and other volatile esters. Taken together, *FaCXE2* and *FaCXE3* are indeed involved in the regulation of the synthesis and degradation of strawberry volatile esters.

## 1. Introduction

Strawberry (*Fragaria* × *ananassa*) is a favorite berry crop for its unique flavor and nutritional value. Studies have shown that esters, furans, terpenes, and sulfur compounds are the substances with the volatile characteristics of strawberries [1,2,3]. Among them, 4-methoxy-2,5-dimethyl-3(2H)-furanone is the characteristic volatile furanone compound in strawberries [4,5,6]. Ester volatiles are the basic aroma components in apples, pears, bananas, strawberries, and other fruits [7,8,9], and esters can reach up to 25–90% of the total content of volatile substances in strawberries [7]. Volatile esters not only contribute to the aroma level of ripe fruits, but also make flowers more attractive to pollinators and protect plants by inducing plant defense pathways. Some common esters, such as ethyl acetate, methyl butyrate, octyl acetate, octyl butyrate, hexyl acetate, benzyl acetate, and hexyl 2-butyrate, were identified in strawberry fruit [10]. The most abundant esters in cultivated strawberries are methyl butyrate, ethyl butyrate, ethyl hexanoate, etc., while the ones that contribute more to the aroma of wild strawberries are butyl formate and octyl acetate [8].

The aroma components of volatile esters come from two precursors, fatty acids and amino acids [11]. Many genes and enzymes are involved in these two metabolic pathways (Figure 1) [12,13,14,15]. Fatty acids and general linoleic and linolenic acids are degraded by oxidative degradation by lipoxygenase (LOX) or hydroperoxide lyase (HPL) into volatile aldehydes, such as hexanal and *Z*-2-hexenal [16,17], which are then converted to alcohols by alcohol dehydrogenase (ADH) [18]. The β-oxidation of fatty acids leads to the formation of C2 units (acetyl-CoA) [19]. In another pathway, the amino acids form aldehydes by aminotransferase (ATF) and decarboxylase (DCX), or directly by aldehyde synthase (ADS) [20]; aldehydes are then converted to alcohols by alcohol dehydrogenase (ADH) [15]. The last important step in the biosynthesis of volatile esters is the esterification of alcohol with acetyl CoA by catalyzing different alcohol acyltransferases (AAT) [21], which is the rate-limiting step (Figure 1) [22].

Carboxylesterases (CXEs, EC 3.1.1.1) are hydrolytic enzymes, which structurally belong to the α/β hydrolase superfamily [21]. These enzymes hydrolyze short-chain fatty acid esters and have shown powerful functions in the animal kingdom, such as neurotransmission in animals [23], insecticide resistance in insects [24], and xenobiotic detoxification in microorganisms [25], and so on, while in the plant kingdom, CXEs are originally used only for isozyme labeling. At present, 20 carboxylesterase genes have been systematically excavated from the genome of *Arabidopsis thaliana* and divided into seven groups [26]. The physiological functions of carboxylesterases in plants have been studied. For example, AtCXE18 and AtCXE12 are responsible for the hydrolysis of exogenous toxic substances such as herbicides [27,28,29], and AtCXE8 has been shown to be involved in the resistance of Arabidopsis to Botrytis cinerea [29]. Some CXE genes are associated with the hydrolysis of volatile esters in plants, such as tomato (SlCXE1) [30], apple (MdCXE1) [31], peach (PpCXE1) [32], and strawberry (FanCXE1) [33]. Hexyl acetate and *E*-2-hexenyl acetate have been identified as substrates for SlCXE1 and PpCXE1. SlCXE1 has a major role in regulating the volatile ester content of tomato fruit, and the relatively high expression of SlCXE1 in tomato fruit leads to low acetate ester contents, which affects the overall aroma of the fruit [30].

In this study, we explore the key carboxylesterase family members (FaCXEs) that affect the catabolism of volatile esters in strawberry from the perspective of bioinformatics analysis, gene expression analysis, and the enzymatic characterization of carboxylesterase family members. Our results indicate that FaCXEs have a role in controlling the esters metabolism in strawberry fruit, which will help improve the overall flavor of popular strawberry.

## 2. Results

### 2.1. Volatile Esters Are the Basic Aroma Components in Some Fruits

The solid-phase microextraction (SPME) was used to extract the volatile aroma components from five fruits, including monocots plant banana, dicots Rosaceae plant strawberry, apple, pear, and peach. Then, we used a high-performance gas chromatography instrument (7890A-5975C) to detect the volatile ester aroma components. The result showed that a total of 57 volatile ester components were detected in five kinds of fruits, and the types of volatile ester aroma components were great differences among the different fruits (Table 1). Hexyl acetate, butyl butyrate, and methyl butyl caproate were detected in the five kinds of fruits. Additionally, *E*-2-hexenyl acetate, hexyl butyrate, butyl acetate, and 2-methyl-1-butanol acetate were detected in four kinds of fruits. The data indicated that volatile esters are the basic aroma components in these five fruits. There are also some compounds which are only observed in strawberry, such as *E*-2-hexenyl butyrate, *Z*-3-hexenyl acetate, octyl acetate, etc. In terms of the content of each compound, the content of these components in banana, strawberry, and apple is much higher than that in pear and peach (Appendix A).

### 2.2. Correlation Analysis of Volatile Ester Components and Related Genes Expression during Strawberry Development

We used GC-MS to detect the changes in the volatile aroma components in strawberry fruits at different maturation stages (Figure 2A,B, Appendix A, Appendix A). The results showed that the content of almost all of the detected esters, except for 2-pentyl acetate compounds, significantly increased and reached the highest level in fruit at the red stage; less at early developmental stages; and the lowest at the middle stages of the fruit’s development. For example, methyl hexanoate, methyl acetate, methyl butyrate, and *E*-2-hexenyl butyrate were significantly detected in strawberry fruit at the maturity stages. Most alcohols did not change as much as the lipids. As the fruit matured, many alcohols showed a slow decrease, such as hexanol and octen-3-ol, whose content gradually decreased with the maturation of the fruit, while linalool showed the opposite trend.

Based on the transcriptomic analysis, the expression of the genes related to the volatile components in strawberry fruits at different maturation stages is shown in Figure 2C. The results showed that except for hydroperoxide lyase (HPL), aldehyde synthase (ADS), and decarboxylase (DCX), the rest of the genes were large families. The peak of the transcript abundance of most genes such as lipoxygenase (LOX) and hydroperoxide lyase (HPL), aminotransferase (ATF) in the amino acid pathway, aldehyde synthase (ADS), and alcohol dehydrogenation (ADH) appeared at the early fruit developmental stages. However, DCX in the amino acid pathway appeared at the late stages of the fruit’s maturation. 

Based on the gene annotation and the conserved sequence GXSXG of the CXE members, a total of 35 CXE members were identified from the strawberry genome website. The expression of only 30 CXE genes was detected in the transcriptome data of strawberry fruit, while the expression of *FaCXE18*, *FaCXE32*, *FaCXE33*, *FaCXE34,* and *FaCXE35* was not detected. The expression pattern of the strawberry CXE genes were also inconsistent in Figure 2C. Some CXE genes showed the highest transcript abundance in fruit at the early stages, some at the later stages, and some at the middle stages.

### 2.3. Identification of Carboxylesterases Involved in the Metabolism of Volatile Ester Compounds in Strawberries 

The CXE members of strawberry, apple [31], peach [34], pear, and banana plants were screened through the genome database in this study. Additionally, the phylogenetic tree was constructed by MEGA6.0 (Figure 3). The tree showed that the CXE members of the above five fruits were divided into seven different groups. It has been proved that the CXE genes, including PpCXE1 of peach [32], SlCXE1 of tomato [30], and FanCXE1 of strawberry [33], which have the function of hydrolyzing volatile esters in fruit, were all located in group 3. That is why the CXE genes in group three are our focus. Figure 3 shows that the CXEs belonging to group 3 contained 17 strawberry members, 15 peach members, 7 apple members, 13 pear members, and 10 banana members. There are 14 genes from the CXE genes of group 3 clustered on strawberry chromosome 2 to form a gene cluster. *FaCXE2* and *FaCXE3*, which had the most abundant transcripts in group 3, were selected for a further cloning, protein expression, and enzymatic analysis.

*FaCXE2* and *FaCXE3* were cloned using strawberry green fruit cDNA as a template. The specific primers are listed in Appendix A. *FaCXE2* and *FaCXE3* were introduced into the heterologous expression vector pRSF-duet and then transferred to the *Escherichia coli* strain BL21 (DE3) for the protein induction and expression. Then, the fusion proteins were purified by affinity chromatography. The SDS-PAGE analysis showed that the molecular weight of the fusion proteins was approximately 35 to 40 KDa, which was basically consistent with the predicted results by ExPASy ProtParam. 

To explore the natural substrates of FaCXE2 and FaCXE3, SPE technology was used to isolate and extract the volatile esters contained in strawberry fruits at the red stage, and then the purified recombinant protein was used for an incubation reaction with these substances. The strawberry volatile esters which were detected were used as the reaction substrates. The contents of some esters such as hexyl acetate, *E*-2-hexenyl acetate and methyl hexanoate, *Z*-3-hexenyl acetate, etc., reduced significantly. However, the contents of the aldehydes, including hexanal and nonanal, were basically unchanged. These results indicate that hexyl acetate, *E*-2-hexenyl acetate and methyl hexanoate, and *Z*-3-hexenyl acetate may be the natural substrates of FaCXE2 and FaCXE3 (Figure 4). 

We further used hexyl acetate, *E*-2-hexenyl acetate, and *Z*-3-hexenyl acetate as substrates, and employed GC-MS to analyze the enzymatic products of rFaCXE2 and rFaCXE3. The corresponding products were detected in each recombinant protein reaction, which hydrolyzed hexyl acetate, *E*-2-hexenyl acetate, and *Z*-3-hexenyl acetate to 1-hexanol, *E*-2-hexenol, and *Z*-3-hexenol (Figure 5). 

To further investigate the hydrolysis activity of the volatile esters of FaCXE2 and FaCXE3, the enzyme kinetics of the recombinant proteins were also analyzed (Table 2). The results showed that the affinity of FaCXE3 for each substrate was much greater than that of FaCXE2. The substrates affinity (*K_m_*) of FaCXE3 for hexyl acetate, *E*-2-hexenyl acetate, and *Z*-3-hexenyl acetate were 0.84, 0.98, and 0.95 mM, respectively. While rFaCXE2 had the highest affinity for *E*-2-Hexenyl acetate (1.04 mM).

### 2.4. Expression Pattern of FaCXEs in Strawberry Fruits

The above results showed that the recombinant FaCXE2 and FaCXE3 had an activity of hydrolyzing acetate esters in vitro. So, which is the key gene involved in the regulation of the volatile esters in fruit? Based on the transcriptome data (Figure 6), the expression levels of some genes were significantly lower than other genes, such as *FaCXE12*, *FaCXE14,* and *FaCXE31*, while the expression levels of some genes such as *FaCXE2* and *FaCXE27* (*FanCXE1*) were higher than others at the last two stages of maturation (T and R). The expression trends of *FaCXE2*, *FaCXE3*, and *FaCXE27* during fruit maturation are also different. The expression of *FaCXE2* gradually increased with the fruit maturation, the expression level of *FaCXE3* showed a decreasing trend, and the expression level of *FaCXE27* decreased firstly and then increased. These results indicate the complexity of the true function of CXEs.

### 2.5. Transient Overexpression of FaCXEs in Strawberry Fruit

To further validate the functional differences of the two *FaCXEs* in vivo, *FaCXE2* and *FaCXE3* were overexpressed by a transient expression technique in strawberry fruits (Figure 7). After the transient overexpression of *FaCXE2* and *FaCXE3*, the contents of *Z*-3-hexenyl acetate, methyl hexanoate, and methyl butyrate were decreased in transgenic fruits relative to the mock controls. When the two genes were compared horizontally, the result showed that the overexpression of *FaCXE3* and the overexpression of *FaCXE2* had a 2–4 folds difference in the reduction in the same ester in the fruit, and the activity of FaCXE3 was higher. 

The above results show that FaCXEs play an important role in the content of volatile esters in strawberry fruit, but only from the perspective of a transient expression. More research is needed for a stable expression.

## 3. Discussion

Volatile esters play a very important role in higher plants. They make flowers more attractive to pollinators and dispersing animals [35,36,37,38], act as protectants against pathogens by inducing several important plant defense pathways [39,40], and contribute to the aroma of ripe fruits [41,42]. These compounds are produced by all berry varieties during the ripening process and play a key role in determining the final organoleptic quality of the fruit. In fruits including apples, pears, and bananas, esters are the main components of their characteristic aroma [12,43,44,45]. However, in strawberries, esters have a mixing effect on the volatiles that constitute the aroma, thereby affecting the quality of the strawberry [46,47].

CXEs in the plant kingdom are regarded as a superfamily of proteins [48]. The research on the physiological functions of the CXE family first began in the model plant *Arabidopsis thaliana* [26]. The CXE family is divided into seven groups, and it appears that the CXEs of each subgroup have different functions. For example, the second group is involved in plant–pathogen interactions and the carboxylesterases involved in the degradation of volatile esters are mostly distributed in the third group [32].

Ester aromatic substances refer to compounds containing ester (-COO-) functional groups in a chemical structure which are formed by the dehydration of hydroxyl (-OH) and carboxyl (-COOH) groups [47,49]. Volatile ester substances are also divided into two types: one is linear esters and the other is branched esters [50]. There are also differences in the composition of ester volatiles in different kinds of fruits. Most of the banana fruits are linear esters, such as butyl acetate and ethyl butyrate and branched esters such as isoamyl acetate [51]. Linear esters such as butyl acetate, ethyl acetate, and hexyl acetate in strawberries are important aroma substances [52]. Our analysis found that hexyl acetate, methyl butyl caproate, and butyl butyrate were detected in all five fruits including strawberry, apple, pear, peach, and banana (Table 1). These results suggest that volatile esters are the basic aroma components.

In different animals or plants, some carboxylesterases have a high specificity, acting only at a very high rate for specific substrates (e.g., acetylcholine) [53], while others can hydrolyze a wide range of substrates [54]. The CXEs involved in regulating the volatile ester contents are being reported, and most of the substrates belong to the acetates. The MdCXE1 recombinant protein could hydrolyze a range of 4-methyl umbelliferyl esters [31]. The functions of SlCXE1 from tomatoes [30] and PpCXE1 from peaches [32] were also confirmed to hydrolyze acetate esters. In addition to acetate esters, the reported strawberry FanCXE1 could also decompose butyrate and hexanoate esters. In this study, we analyzed the CXE family of strawberry and found that the CXEs in group 3 were clustered on the chromosomes. When we used strawberry extract as a substrate, the results showed that the FaCXEs of strawberry were not only limited to hydrolyze acetate esters, but also had the ability to hydrolyze other types of esters, such as methyl hexanoate, 2-hexenyl butyrate, etc. (Figure 4). The function of FaCXEs were similar to that of FanCXE1, as previously reported [33].

Exploring the regulation mechanism of the volatile esters in strawberry fruit from the synthesis and hydrolysis provides a more comprehensive perspective for understanding the formation and regulation of the fruit quality and provides an important theoretical basis for improving the aromatic quality of strawberry fruit by molecular breeding technology. In the future, we will further investigate the effect of these CXE genes on the flavor quality of strawberry fruit by an RNA interference (RNAi).

## 4. Materials and Methods

### 4.1. Material

The strawberry (*Fragaria* × *ananassa*) fruits, including Christmas Red, Wanxiang, Suizhu, etc., in this study were collected from the Strawberry Germplasm Resource Garden of Anhui Agricultural University. Banana (*Musa acuminata*), apple (*Malus domestica*), pear (*Pyrus bretschneideri*), and peach (*Prunus persica*) samples were purchased from the market. Benihoppe fruit samples at different maturation stages, including small green fruit (SG), mid-sized green fruit (MG), big green fruit (BG), white fruit (W), turning fruit (T), and red fruit (R), were harvested at 9, 14, 18, 21, 24, and 30 days after flowering, respectively. 

### 4.2. Preparation and Detection of Fruit Volatile Extracts

The volatile aroma compounds from the fruits, including strawberry, banana, apple, pear, and peach, were analyzed by headspace solid-phase microextraction (SPME). First, the five fruits, including strawberry, banana, apple, pear, and peach, were ground into powder in liquid nitrogen, respectively. Subsequently, 6 g of power was poured into a 100 mL headspace solid phase bottle and then 50 μL of cyclohexanol was added as the internal standard and 2 g of NaCl was added to promote the full volatilization of the substances. The mixtures were incubated at 50 °C for 10 min. The volatiles were collected by a 50/30 μm polydimethylsiloxane and divinylbenzene (DVB/CARBOXEN-PDMS) fiber, which was aged for 2 h at 250 °C, then the needle was inserted into the headspace solid phase bottle and extracted at 50 °C for 50 min. The extraction fiber was desorbed in a gas chromatograph injection port at 250 °C for 5 min (Aglient GC-MS 7890A-5975C, Agilent, CA, USA).

The mass spectrometry conditions: the chromatographic column was a DB-5MS capillary column, the carrier gas was helium (He) (purity 99.999%), the flow rate was 1.2 mL·min^−1^, and the sample port temperature was 250 °C. The temperature program: the initial temperature was 40 °C and held for 3 min and raised to 100 °C at 3 °C·min^−1^ and held for 3 min, and then to 245 °C at 5 °C·min^−1^. The injection method was in splitless mode.

The calculation method: ① the mass of the internal standard was calculated as follows: m1 = internal standard density × internal standard volume = 0.9624 g/mL × 0.05 mL = 0.04812 g; ② the mass of the internal standard was added in each independent experiment: m2 = m1·dilution factor = 0.04812 g 5 × 10^−4^ = 2.406 × 10^−5^ g; ③ the mass of the product was calculated: mx = m2·product peak area/internal standard peak area; and ④ the relative mass of the product was obtained: m = mx/powder mass = mx/6 g.

### 4.3. Bioinformatic Analysis of CXE Genes Family

The CXE genes of strawberry and apple were identified by Souleyre et al. [31]. Peach CXE genes were identified previously by Zhang et al. [34]. Pear and banana CXE genes were searched from the National Center for Biotechnology Information database (NCBI, https://www.ncbi.nlm.nih.gov/) (accessed on 10 September 2022). 

The phylogenetic tree was constructed using MEGA 6.0 software by a neighbor-joining method with 1000 bootstrap replications. The accession numbers of these CXE genes were listed in Appendix A. The molecular weight of the FaCXEs were predicted by the ExPASy ProtParam (http://web.expasy.org/protparam/) (accessed on 25 November 2020) [55].

### 4.4. Expression Analysis of CXE Genes

The total RNA from strawberry fruits was isolated using an RNAprep Pure Plant Kit (Polysaccharides and Polyphenolics-rich) (TIANGEN, Beijing, China) according to the manufacture’s protocol. The quantity of the total RNA was measured on an SMA 4000 UV-VIS spectrophotometer (Merinton, Beijing, China), and the quality of the total RNA was detected by 1.2% gel electrophoresis. The first-strand cDNA was synthesized according to the manufacture’s instruction by using a PrimeScript RT Reagent Kit (Takara, Dalian, China). The obtained cDNA was stored at −20 °C until its use. Strawberry fruits at different developmental stages were collected and their RNA was extracted. cDNA libraries for the DNBSEQ platform at BGI were constructed by following the method previously described [56]. The FPKM values of the transcripts were calculated to analyze the expression level of the genes at different fruit stages. RT-qPCR was performed using SYBR Green Supermix Kit on a CFX96 instrument (Bio-Rad, CA, USA). A real-time quantitative PCR (RT-qPCR) reaction system with a volume of 20 μL consisted of 10 μL of SYBR Green Supermix, 0.8 μL specific primers (10 μM), 200 ng cDNA templates of strawberry fruits at different stages, and RNase free-H2O. The reaction program for the qPCR was: 95 °C for 3 min, 40 cycles of 95 °C for 30 s, 60 °C for 30 s, and 72 °C for 20 s, and then a melting curve analysis from 65 to 95 °C. Three biological replicates for different stages in strawberry fruits were analyzed. The primers used for the RT-qPCR are listed in Appendix A. 

### 4.5. Cloning of FaCXE Genes

The specific primers (Appendix A) were designed to amplify the open reading frames (ORFs) of *FaCXEs* using Phusion high-fidelity polymerase (Thermo Scientific, Vilnius, Lithuania) following the manufacture’s protocol. The PCR program was as follows: 98 °C for 3 min, followed by 30 cycles of 98 °C for 30 s, 62 °C for 30 s, 72 °C for 1 min, and then an extension at 72 °C for 10 min. The PCR products were purified from gel and then introduced into a pEASY blunt-simple vector (Transgen, Beijing, China) for sequencing. 

### 4.6. Construction of Prokaryotic Expression Vector and Expression of Recombinant Proteins

The ORFs of the *FaCXEs* were inserted into the pRSF-Duet expression vector with 6x His-tag, and then transferred into the *Escherichia coli* strain BL21 (Transgen, Beijing, China). The empty vector pRSF-Duet, used as the control, was also transferred into the strain BL21. 

The positive cells with *FaCXEs* and the empty vector were picked, respectively, and then cultured in 200 mL of a Luria–Bertani (LB) liquid medium containing 50 mg/L of kanamycin in a 37 °C incubator at 220 rpm. When the density of the bacterial cells approximately reached 0.6 at 600 nm, 180 μL of isopropyl β-D-thiogalactopyranoside (IPTG) (1 M) was added to the cultures. After the induction of the culture for 24 h at 16 °C, the cells were collected by centrifugation at 4 °C at 6000× *g* for 10 min. Recombinant CXE proteins and His protein were purified by affinity chromatography. The molecular weight of the purified proteins was analyzed by SDS-PAGE.

The recombinant CXEs activities and kinetics. The detection of the enzyme’s activities was carried out using strawberry aroma extracts as the substrate. The fruit volatiles were extracted by a solid phase extraction (SPE) method. Accurately, 6 g of powdered fruit, which was freeze-dried, was transferred to a conical flask with a stopper and 160 mL of pure water at 100 °C was added to incubate for 10 min, and then the fruit extracts were filtered out. After cooling, the extracts were transferred to a separatory funnel, then 40 mL of extraction solvents (80% pentane and 20% MTBE) was added and homogenized. After standing, the supernatant was pipetted into a tube. The remaining residues were repeatedly extracted twice. These supernatants were centrifuged at 10,000× *g* for 10 min and then pooled together in a new tube. The extracts were dried by adding anhydrous sodium sulfate and concentrated to 1 mL using a nitrogen blower. The concentrated extracts were passed through an SPE column to remove the impurities. The concentrated extracts were slowly added through the SPE column, which was firstly activated with 6 mL of the eluents (50% pentane and 50% MTBE). Then, the flask containing the concentrates was repeatedly rinsed with 10 mL of the eluents and the mixtures were added through the SPE column again. The solution was collected and concentrated to 200 μL and then sealed until use. 

The FaCXEs activity assays were conducted in a 400 μL enzyme reaction system which consisted of 20 μL of strawberry aroma extracts, 20 μg of purified protein, and 100 mM of Tris-HCl buffer (pH 7.5). The reaction mixtures were incubated at 30 °C in a water bath for 30 min, then a 50/30 μm DVB/CARBOXEN-PDMS extraction needle was inserted into the headspace of the solid phase bottle. After extraction for 30 min, the reaction products were identified by GC-MS. The detection method of the GC-MS was the same as the above method. 

The enzyme activity analyses of the recombinant FaCXE proteins were performed using hexyl acetate, *Z*-3-hexenyl acetate, and *E*-2-hexenyl acetate as the substrates. A 400 μL reaction mix which contained 2 μL of 10 mM volatile ester substrates (hexyl acetate, *Z*-3-hexenyl acetate, and *E*-2-hexenyl acetate), 10 μg of purified protein, and 100 mM of Tris-HCl buffer (pH 7.5). The mixtures were incubated at 30 °C for 30 min and were then extracted with the same volume of hexane. The extracts were detected using GC-MS. 

Enzyme kinetic analysis. The enzyme reaction rates were analyzed at different substrate concentrations. The contents of the different products were estimated using standard curves. The *K_m_* values for the different substrates were calculated using double-reciprocal curves.

### 4.7. Construction of Binary Vectors and Transient Overexpression in Strawberry Fruits 

The ORFs of FaCXE2 and FaCXE3 were constructed into the pCB2004 binary vector by the Gateway Cloning system (Invitrogen, CA, USA), as reported previously. Briefly, the *FaCXEs* with an attB linker sequence were cloned into the entry vectors pDONR207. After the sequence confirmation, the Gateway vectors pCB2004 were used to construct the binary vectors, respectively, named pCB2004-*FaCXE2* and pCB2004-*FaCXE3*. *FaCXE2* and *FaCXE3* were driven by the 35S promoter. Subsequently, these expression vectors and pCB2004 empty vector were transformed into *A. tumefaciens* (GV3101) by electroporation. The transient overexpression of strawberry was performed using the positive GV3101 strain carrying the pCB2004-*FaCXEs* plasmids. The GV3101 containing the pCB2004 empty vector was used as a control.

The transient transformation of strawberries. The strawberry fruits at the white stage were injected by *A. tumefaciens* carrying a pCB2004-*FaCXE2*, pCB2004-*FaCXE3,* and pCB2004 empty vector, respectively. The *A. tumefaciens* suspension was injected into the fruits from the strawberry pedicle. Two to four days after the injection, these fruits were harvested and stored at −80 °C for the analysis of the volatile aroma’s components. More than 10 strawberry fruits were used in the detection of volatiles. Three biological replicates were completed in this assay.

## Figures and Tables

**Figure 1 ijms-24-00383-f001:**
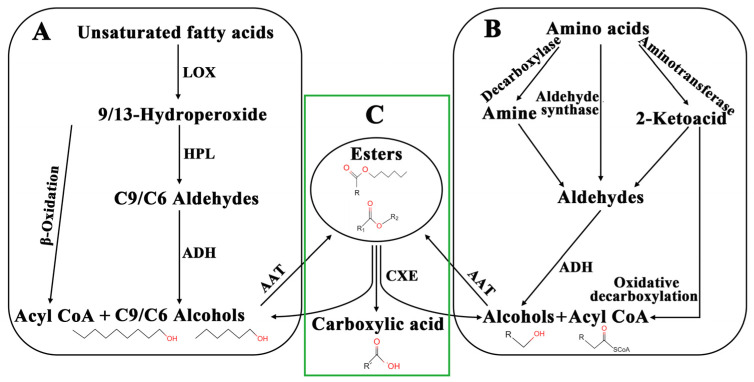
Metabolic pathway of volatile esters in strawberry. (**A**) Fatty acid pathway. (**B**) Amino acid pathway. (**C**) CXE pathway. LOX, lipoxygenase; HPL, hydroperoxide lyase; ADH, alcohol dehydrogenase; AAT, alcohol acyltransferases.

**Figure 2 ijms-24-00383-f002:**
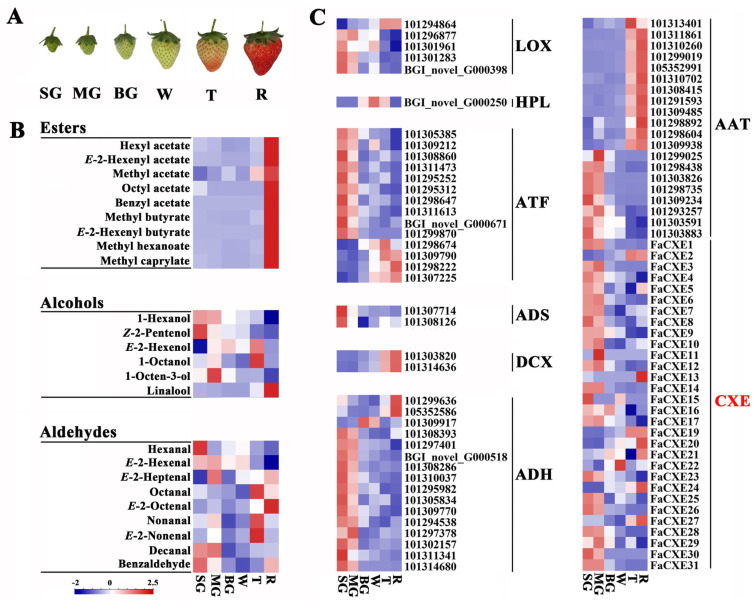
Accumulation pattern of substances and expression of genes involved in the metabolic pathway of volatile esters in strawberry fruits at different maturation stages. (**A**) Strawberry fruits at different maturation stages. (**B**) Heatmap in the contents of esters, alcohols and aldehydes at different maturation stages. (**C**) The expression pattern of genes involved in the metabolic pathway of aroma substances in strawberry fruits at different maturation stages.

**Figure 3 ijms-24-00383-f003:**
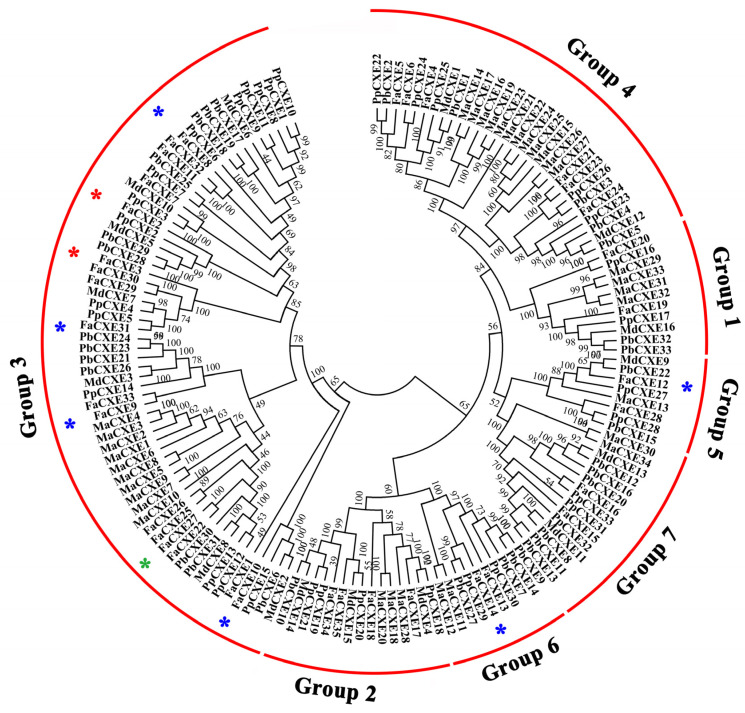
Phylogenetic tree of CXE genes from the five plants. Fa represents strawberry; Ma represents banana; Md represents apple; Pb represents pear; and Pp represents peach. The phylogenetic tree was constructed using the full-length amino acid sequences of selected CXEs by NJ methods. All CXEs from the five plants, including strawberry, banana, apple, pear, and peach were classified into 7 groups. The function of FaCXEs, marked with red asterisks, were identified in this study. The function of FaCXE27 (FanCXE1), marked with green asterisk, was identified in earlier study [33]. Red, green and blue asterisks denote the FaCXE genes selected for expression.

**Figure 4 ijms-24-00383-f004:**
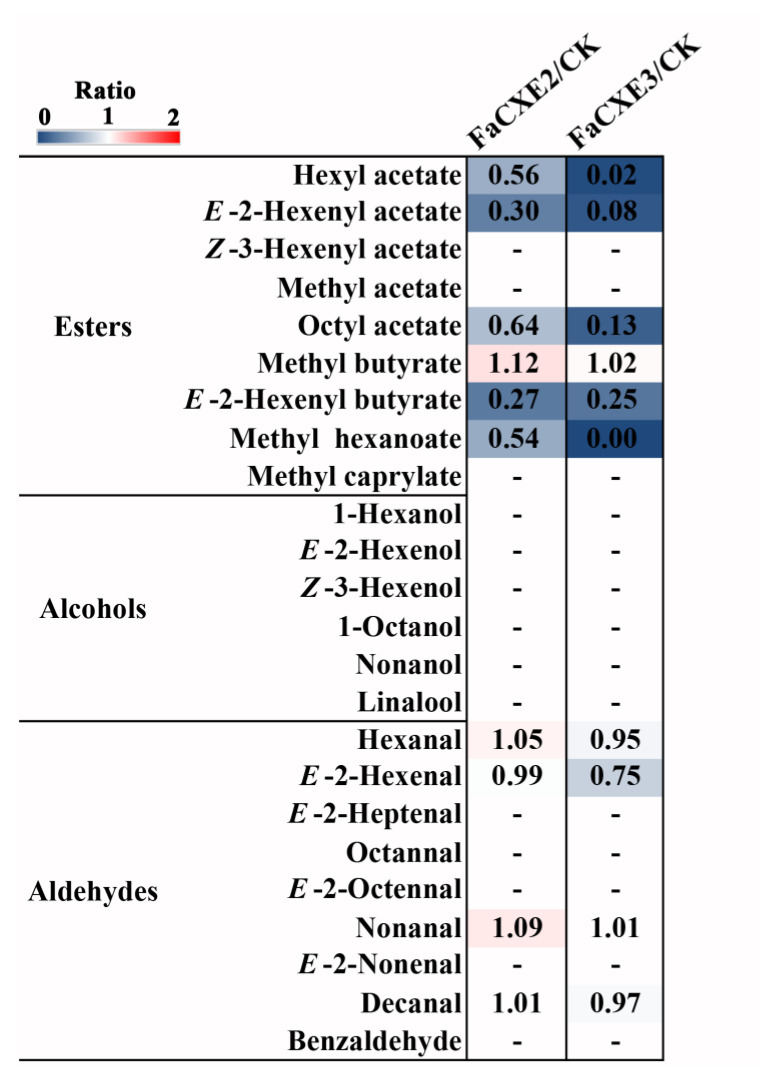
Hydrolysis analysis of volatile substances extracted from strawberry fruit by recombinant FaCXE2 and FaCXE3 proteins. The volatile compounds extracted from strawberry red fruit (SPE) were used as the substrate, and the recombinant FaCXEs were used as the enzyme. The controls (CK) were treated with boiled recombinant protein. Numbers represent ratio of volatiles in the hydrolysis assays compared with the controls, “-” means not detected. All data are presented as mean of three biological replicates.

**Figure 5 ijms-24-00383-f005:**
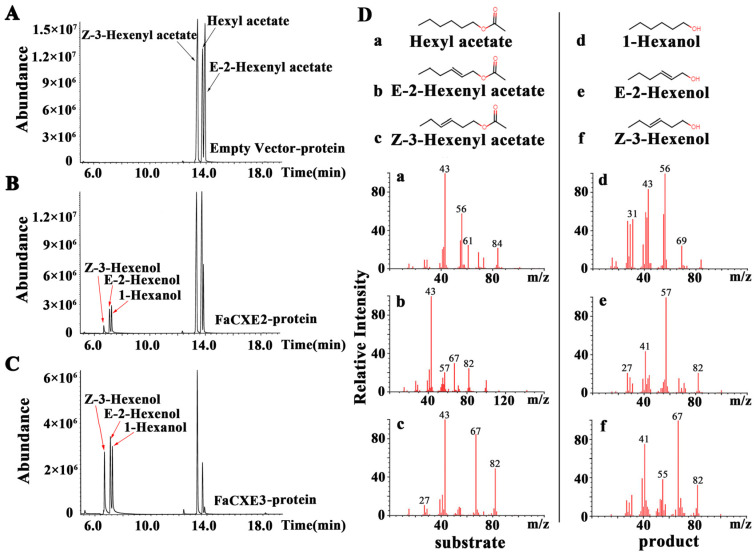
Enzyme activity analysis of rFaCXE2 and rFaCXE3. (**A**–**C**) The hydrolysis activities of FaCXEs using hexyl acetate, *E*-hexenyl acetate, and *Z*-3-hexenyl acetate as substrates. (**D**) Chemistry structural formulas and MS/MS profiles of three substrates and three products.

**Figure 6 ijms-24-00383-f006:**
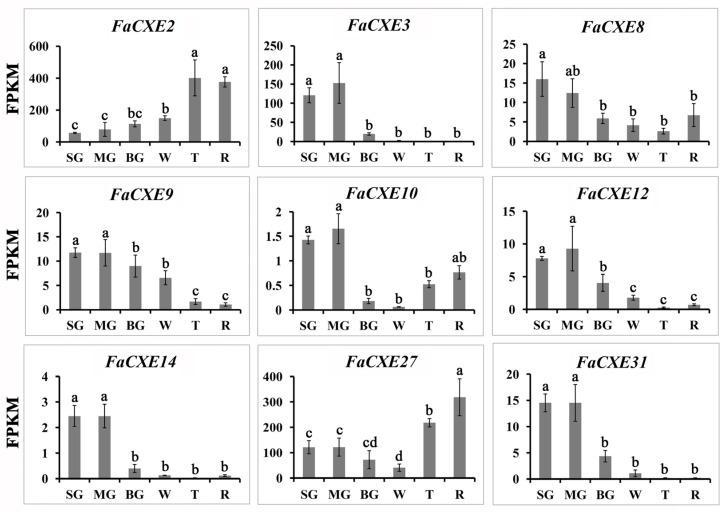
Expression levels of *FaCXEs* in strawberry fruits at different maturation stages. All data are presented as mean of three biological replicates and each error bar indicates ± SE. Lowercase letters indicate significant differences at *p* < 0.05.

**Figure 7 ijms-24-00383-f007:**
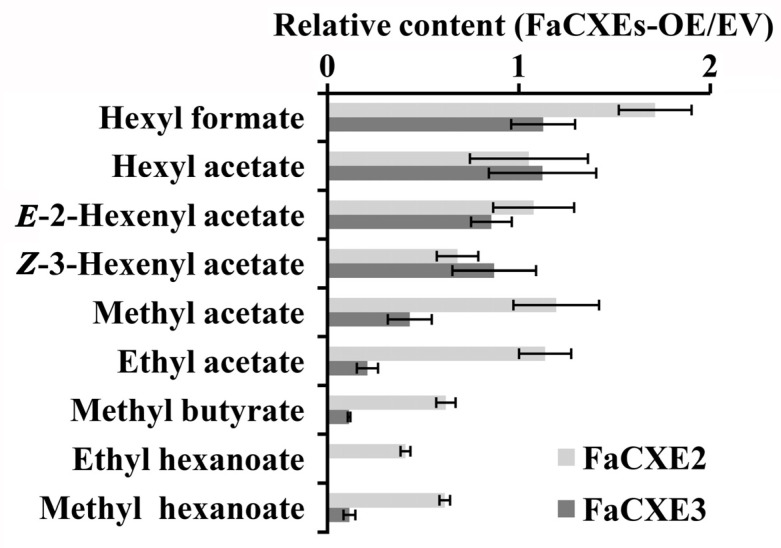
Relative content analysis of volatile esters in transiently overexpressing *FaCXEs* strawberry fruits and empty vector (EV). In all biological replicates, at least 10 fruits were used for used for each construct. Data are presented as mean ± SE obtained from three biological replicates.

**Table 1 ijms-24-00383-t001:** Content of volatile ester compounds in five kinds of fruits (ng g^−1^ FW).

	Volatile Esters	Fa	Ma	Md	Pb	Pp
1	Hexyl acetate	4270	2940	2740	39.1	166
2	Butyl butyrate	1050	5060	277	38.4	69.4
3	*E*-2-hexenyl acetate	1110	241	-	11.8	72.4
4	Hexyl butyrate	2550	7600	693	15.3	-
5	Butyl acetate	940	1720	1680	39.1	-
6	Methyl butyl caproate	229	525	52.3	5.82	19.5
7	2-methyl-1-butanol acetate	119	8070	4360	17.3	-
8	Butyl isovalerate	76.3	26,100	63.6	-	-
9	*E*-2-hexenyl butyrate	1030	-	-	-	-
10	Octyl acetate	25,500	-	-	-	-
11	Methyl hexanoate	390	102	-	26.3	-
12	Butyl acrylate	-	-	30.2	53.1	64.7
13	Isobutyl acetate	143	1040	155	-	-
14	Ethyl hexanoate	5230	-	-	1380	3350
15	Heptyl acetate	328	11,500	-	27.9	-
16	Methyl acetate	-	-	-	146	-
17	Ethyl acetate	613	271	-	23.1	-
18	Ethyl propanoate	59	-	-	-	-
19	n-Propyl acetate	65.8	-	184	-	-
20	Methyl butyrate	1470	-	-	14.3	-
21	Methyl 2-methyl butyrate	43.6	-	-	-	-
22	Ethyl butyrate	1760	-	-	-	-
23	Propyl propionate	-	-	178	-	-
24	Isopropyl butyrate	16.4	-	-	-	-
25	2-pentyl acetate	-	7170	-	-	-
26	Propyl butyrate	79.8	-	284	-	-
27	Amyl acetate	-	-	283	-	-
28	2-Methylbut-2-en-1-yl acetate	135	-	22	-	-
29	Propyl 2-methylbutyrate	-	635	160	-	-
30	2-Methylbutyl isobutyrate	-	40.4	-	-	-
31	Propyl 2-methyl butanoate	-	2470	-	-	-
32	2-Pentanyl propanoate	-	-	78.3	-	-
33	Isopentyl butyrate	-	159	-	-	-
34	*Z*-3-hexenyl acetate	455	-	-	-	-
35	Isopentyl butyrate	-	7380	-	-	-
36	Ethyl butyrate	-	654	-	-	-
37	Hexyl 2-methylbutyrate	-	-	281	-	-
38	Hexyl 3-methylbutyrate	-	471	-	-	-
39	Isoamyl 2-methylbutyrate	-	505	-	-	-
40	3-methylbut-2-enyl butanoate	-	228	-	-	-
41	Isoamyl isovalerate	-	5420	128	-	-
42	Methyl caprylate	2420	-	-	-	-
43	Ethyl 3-methylvalerate	-	482	-	-	-
44	Methyl phenylacetate	7620	-	-	-	-
45	Methyl salicylate	-	-	-	10.9	87.8
46	Ethyl caprylate	4020	-	-	-	-
47	Nonyl acetate	3550	-	-	-	-
48	Hexyl 3-methyl butanoate	-	1300	-	-	-
49	Pentyl hexanoate	-	-	22.7	-	-
50	Benzyl butyrate	320	-	-	-	-
51	Ethyl *E*-2-decenoate	220	-	-	-	-
52	Octyl butyrate	1400	-	-	-	-
53	Ethyl decanoate	991	-	-	-	-
54	Decyl acetate	7290	-	-	-	-
55	2-Undecyl acetate	1280	-	-	-	-
56	Cinnamyl acetate	797	-	-	-	-
57	Hexadecyl decanoate	625	-	-	-	-

“-” means not detected. Fa represents strawberry; Ma represents banana; Md represents apple; Pb represents pear; and Pp represents peach. The data are presented as mean of three duplicates in the table.

**Table 2 ijms-24-00383-t002:** Enzymatic kinetic analysis of FaCXEs recombinant proteins.

	rFaCXE2			rFaCXE3		
Substrates	*K_m_* (mM)	V_max_ (pKat μg^−1^)	V_max_/*K_m_* (pKat μg^−1^ mM^−1^)	*K_m_* (mM)	V_max_ (pKat μg^−1^)	V_max_/*K_m_* (pKat μg^−1^ mM^−1^)
Hexyl acetate	10.93 ± 0.84	129.00 ± 7.11	11.80 ± 1.04	0.84 ± 0.07	20.10 ± 2.35	24.00 ± 1.75
*E*-2-Hexenyl acetate	1.04 ± 0.05	5.46 ± 0.47	5.24 ± 0.22	0.98 ± 0.11	13.60 ± 1.21	13.80 ± 1.53
*Z*-3-Hexenyl acetate	2.50 ± 0.13	12.50 ± 1.21	5.00 ± 0.32	0.95 ± 0.05	18.50 ± 1.69	19.50 ± 2.03

The data are presented as mean ± SE.

## Data Availability

Data available from the author.

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
