# Peer review of "The Functional Characterization of Carboxylesterases Involved in the Degradation of Volatile Esters Produced in Strawberry Fruits"

_ijms, 2022, doi:10.3390/ijms24010383_

Round 1

Reviewer 1 Report

none

Author Response

Are all the cited references relevant to the research? Can be improved

Response: We have revised the related references.

Reviewer 2 Report

The authors is this study tried to explore the relationship between volatile ester compounds and carboxylesterases (CXEs) during strawberry development. The authors used transcriptomic data to identify CXEs as genes of interest and targeted FaCXE2 and FaCXE3 for further research, including cloning, in vitro enzymatic assays and some in planta experimetation.

Some comments/ points of concern about the manuscript

line 73 ''eaters'' meaning esters?

lines 88 and 92 ''alcohol dehydrogenase'' and ''ethanol dehydrogenase'' are refer to the same enzyme or similar enzymes?

line 99 , while...

line 124 why focused only on esters components, why not other volatile molecules? please explain

line 130 ''datas''

line 130-131 ''The datas indicated that volatile esters are the 130 characteristic aroma components in these five fruits.''

i don't think these data are enough to draw that conclusion

please rephrase

line 140 ''of almost the detected esters...'' almost what? almost all? difficult to understand, please correct

line 144-145 why detected at maturity, i see they where detected in all stages at various concentrations

line 147-148 the authors mention alcohols and aldehydes together making the conclusions confusing, please rephrase

line 150-151 ''statistically shown'' ? what does that mean. what kind of statistics did the authors use?

line 171 it has been proved by whom? please cite the published work

line 184 SDS-PAGE not SDS-PAGE

line 186 predicted results by whom? please explain

line 195 products or substrates?

line 324 boiled not boiling

Some general comments

The Figures need to incluse statistical analysis of the data, error bars, appropriate statistical tests etc

Did the authors used an internal control for the efficiency of the transformation of the strawberries? please explain

Reviewer 3 Report

This manuscript is interesting, it is written clearly and concisely, and it provides reliable and useful information for a better understanding the formation and regulation of fruit quality, and provides an important theoretical basis for improving the aromatic quality of strawberry fruit by molecular breeding technology.

However, some observations that should be addressed by the authors are indicated below.

On line 37 it says: “…volatile esters were significantly accumulated with the ripening of strawberry fruits.”. I think it would be better to say: “…volatile esters were significantly accumulated with the maturation of strawberry fruits.”. This is because the strawberry fruit is considered a non-climacteric fruit.

On line 45 it says: “…and the least during fruit development.”. I think it would be better to say: “…and the least during the following maturation stages.”.

On line 73 it says: “…, and eaters…”. I think it should say: “…, and esters…”.

On line 106 it says: “…Some CXE genes is associated with…”. I think it would be better to say: “…Some CXE genes are associated with…”.

On line 121 it says: “…The solid-phase microextraction (SPME) were used to extract…”. I think it would be better to say: “…The solid-phase microextraction (SPME) was used to extract…”.

On line 130 it says: “…The datas indicated…”. I think it would be better to say: “…The data indicated…”.

On lines 141-142 it says: “…and reached the highest level in fruit at the maturity stages; …”. I think this statement is not clear because it does not specify the state of maturity in which it occurs the highest level of content.

On line 145 it says: ”… As the fruit ripening, …”. I think it would be better to say: ”… As the fruit matures, …”.

On line 147 it says: “…gradually decreased with the development of the fruit; …”. I think it would be better to say: “…gradually decreased with the maturation of the fruit; …”.

On line 50 it says: “…in strawberry fruits at different developmental stages…”. I think it would be better to say: “…in strawberry fruits at different maturation stages…”.

On line 157 it says “…appeared at the late stages of fruit development.”. I think it would be better to say: “…appeared at the late stages of fruit maturation.”.

On line 209 it says: “…esters in vitro. So which is the key…”. .”. I think it would be better to say: “…esters in vitro. So, which is the key…”.

On lines 210-213 it says: “…Based on transcriptome data (Figure 6) the expression levels of some genes were significantly lower than other genes such as FaCXE10, FaCXE12 and FaCXE14, while the expression levels of some genes such as FaCXE2, FaCXE3, and FaCXE27 (FanCXE1) were higher than others in the fruit.”. However, based on what can be seen in Figure 6, this statement would be unclear and clearly incorrect since on the one hand it would have to be specified that it refers only to the last two stages of maturation (T y R), and on the other, it should not include the genes FaCXE10 y FaCXE3.

Apparently, Figure 1 does not indicate the support of any source or reference.

On line 477 it says: “…the five fruits were ground into powder in liquid nitrogen, respectively.…”. I consider that this statement is not clear, since it does not indicate which five fruits it refers to, nor does it indicate what the word refers to respectively.

On line 544 it says: “…collected by centrifugation at 4 °C at 6000 rpm for 10 min.”. In this case I consider that the centrifugation units should be expressed in g units. This observation also applies to what is indicated on line 556.

Finally, it seems that reference 51 (Li, B. D., et al., RSEM: accurate transcript quantification from RNA-Seq data with or without a reference genome. BMC Bioinf. 2011, 12, 323.) is not cited in the manuscript.

Round 2

Reviewer 2 Report

After the revisions suggested by the reviewers, the manuscript has been significantly improved. The authors addressed successfully my major concerns including the missing references, missing methods, and the internal control in the transformation experiments. They also corrected minor spelling or other mistakes in the text improving its readability.